# HLA-A01 and HLA-B27 Supertypes, but Not HLA Homozygocity, Correlate with Clinical Outcome among Patients with Non-Small Cell Lung Cancer Treated with Pembrolizumab in Combination with Chemotherapy

**DOI:** 10.3390/cancers16173102

**Published:** 2024-09-07

**Authors:** Afaf Abed, Anna Reid, Ngie Law, Michael Millward, Elin S. Gray

**Affiliations:** 1Centre for Precision Health, Edith Cowan University, Joondalup, WA 6027, Australia; anna.reid@ecu.edu.au; 2School of Medical and Health Sciences, Edith Cowan University, Joondalup, WA 6027, Australia; 3School of Medicine, University of Western Australia, Crawley, WA 6009, Australia; michael.millward@uwa.edu.au; 4Department of Medical Oncology, Sir Charles Gairdner Hospital, Nedlands, WA 6009, Australia; ngie.law@health.wa.gov.au

**Keywords:** genomic HLA, NSCLC, chemotherapy, immunotherapy, biomarkers

## Abstract

**Simple Summary:**

Treatment personalization is essential in the era of precision medicine. Biomarkers are necessary to predict clinical outcomes and guide treatment selection to improve efficacy and reduce toxicity. Genomic human leukocyte antigen-class I (HLA-I) has been shown to correlate with outcomes among non-small cell lung cancer (NSCLC) patients treated with immunotherapy. Here, we investigate whether the HLA genotype is predictive of outcomes in regard to the combination of immunotherapy with chemotherapy. We found that the HLA-I supertypes HLA-A01 and -A03, but not HLA homozygosity, overall correlated with survival in NSCLC patients treated with immunochemotherapy. While further validation is needed, these results highlight the potential and variability of the HLA genotyping as a biomarker for immunotherapy delivery.

**Abstract:**

Introduction: Maximal heterozygosity on the human leukocyte antigen (HLA) loci has been found to be associated with improved survival and development of immune-related adverse events (irAEs) among NSCLC patients treated with immunotherapy. Here, we investigated the effect of germline HLA-I/-II on clinical outcomes among NSCLC patients treated with first-line pembrolizumab in combination with chemotherapy. Method: We prospectively recruited patients with NSCLC who were commencing first-line pembrolizumab in combination with chemotherapy. DNA from white blood cells was used for high-resolution HLA-I/II typing. Results: Of the 65 patients recruited, 53 complied with the inclusion criteria. We did not find an association between HLA-I/-II homozygosity and clinical outcome among the studied population. However, the presence of HLA-A01 was associated with unfavourable PFS (HR = 2.32, 95%CI 1.13–4.77, *p* = 0.022) and worsening OS (HR = 2.86, 95%CI 1.06–7.70, *p* = 0.038). The presence of HLA-B27 was associated with improved PFS (HR = 0.35, 95%CI 0.18–0.71, *p* = 0.004) and trends toward improving OS. None of the HLA-I supertypes were associated with the development or worsening of irAEs. Conclusions: The absence of association between genomic HLA-I/-II homozygosity and clinical outcome among patients with advanced NSCLC treated with pembrolizumab in combination with chemotherapy might reflect a diminished role for HLA molecules among patients with low or no PD-L1. HLA-A01 and HLA-B27 might have a role in predicting clinical outcomes among this cohort of patients. Further studies are needed to explore biomarkers for this group of patients.

## 1. Introduction

Lung cancer is the leading cause of cancer-related deaths globally [1]. Although the introduction of immune checkpoint inhibitors revolutionised the treatment of locally advanced or metastatic non-small cell lung cancer (NSCLC), the response rate to single-agent anti-PD1 is only 45% in the first-line setting and predominantly occurs in patients with tumours expressing high levels of programmed death ligand-1 (PD-L1) ≥ 50% [2]. However, only 23–28% of NSCLC patients express PD-L1 in 50% or more of their tumour cells [3,4]. The addition of immunotherapy to chemotherapy improved survival of patients with advanced or metastatic NSCLC, including those with PD-L1 expression in less than 50% of their cancer cells [5]. Therefore, investigating biomarkers to predict who will benefit the most from the combination of chemotherapy and immunotherapy is important in regard to personalised treatment and avoiding chemotherapy when possible in order to reduce toxicity.

Human leukocyte antigen (HLA) is a molecule expressed on somatic and immune cells. There are two main classes of HLA, HLA class I and HLA class II [6]. HLA-I is expressed on all nucleated cells, including somatic, cancer and immune cells, and it presents cell peptides to activate CD8+T cells. HLA-II, on the other hand, is expressed on antigen-presenting cells and mediates the activation of CD4+ helper or regulatory T cells.

HLA plays a major role in tumour antigen presentation and the development of the antitumor immune response [7]. Therefore, low expression or low diversity of HLA molecules will result in a reduced number and variety of antigens presented to T-cells to mount an immune response. Many studies showed that there is a correlation between expressed HLA-I on cancer cells and response and survival among patients treated with immunotherapy [8,9]. In addition, it has been postulated that HLA genetic diversity may influence a patient’s likelihood of response to immune checkpoint inhibitors. Indeed, previous work by our group indicated that genomic HLA-I homozygosity has been associated with worse survival among NSCLC patients treated with single-agent immunotherapy [10,11]. Additionally, HLA-I homozygosity and specific HLA-II alleles have been associated with the occurrence of immune-related adverse events (irAEs) among patients treated with single-agent immunotherapy [12,13].

The HLA locus is encoded by multiple alleles that can be grouped into supertypes based on their peptide-binding groove [14]. The B and F pocket in the peptide-binding groove will determine the characteristics of the amino acids attached to it [14]. The correlation between HLA supertypes and clinical outcomes has been examined across many cancer types besides lung cancer [10,11,15]. The presence of HLA-A03 was reported to be associated with worsening survival among patients with solid organ malignancies treated with immunotherapy alone [15]. This correlation did not reach statistically significant results in the subgroup of patients with the NSCLC cohort. We previously showed that the presence of the HLA-A02 supertype was associated with improved OS while the presence of HLA-A03 was associated with increased risk of irAE [10,12]. However, no study to date has evaluated these associations in patients treated with immunotherapy in combination with chemotherapy. Here, we investigated the association between HLA-I/II homozygosity and HLA-I supertypes and clinical outcome variables, including response, progression-free survival (PFS), overall survival (OS) and irAEs, among patients with locally advanced or metastatic NSCLC treated with pembrolizumab in combination with chemotherapy.

## 2. Materials and Methods

We prospectively collected blood from 65 patients with locally advanced or metastatic NSCLC who planned to be treated with pembrolizumab in combination with chemotherapy in the first-line setting. Patients were recruited from four hospitals in Perth, Western Australia, between June 2018 and July 2021. All procedures were approved by the Human Research Ethics Committees at Edith Cowan University (ECU) (No. 18957) and Sir Charles Gairdner Hospital (No. 2013-246 and RGS0000003289) in compliance with the Declaration of Helsinki. All participants provided written informed consent. Information about patients’ demographics, pre-treatment biochemistry, cancer molecular profile, PD-L1 status, development of immune-related adverse events (irAEs) and survival was collected. Results for PD-L1 status was obtained through performing immunohistochemistry using the Dako 22C3 clone for all patients at PathWest Pathology Services as per the standard diagnostic services and reported as the tumour proportion score (TPS).

Progression-free survival (PFS) represented the time between the start of treatment and disease progression. Overall survival (OS) represented the time between the start of treatment and death. Patients who died due to reasons not related to their NSCLC and those who were lost to follow up were censored. The clinical benefit rate (CBR) represents the ratio of those who had complete response (CR), partial response (PR) or stable disease (SD) for 6 months or more over the total.

DNA was extracted from blood using a QIAamp DNA Blood Maxi Kit (Qiagen, Hilden, Germany). Extracted DNA was used for high-resolution HLA typing at the Institute for Immunology and Infectious disease (IIID) at Murdoch University, as previously described by [16]. HLA-I alleles were grouped based on their peptide-binding repertoire into supertypes [14]. HLA-A and -B supertypes are the only well characterised supertypes.

For statistical analysis, the data were dichotomised based on the homozygosity at one or more HLA-I/II loci, maximal heterozygosity at all loci, or based on the predicted HLA-supertypes. PFS and OS associations with HLA-I/II zygosity and HLA -supertypes were evaluated using the log-rank (Mantel–Cox) test and Kaplan–Meier plots were generated using GraphPad Prism V.8 (GraphPad Software, Inc., San Diego, CA, USA). The rate of HLA-I/II zygosity was compared between responders and non-responders and with irAE classifiers using a two-tailed Fisher’s exact test.

We conducted univariate and multivariate analyses using Cox regression models in IBM SPSS Statistics, Version 28.0 (Armonk, NY, USA), comparing OS based on HLA homozygosity and controlling for tumour PD-L1 expression, pre-treatment NLR, age, ECOG status, smoking and sex. Patients with missing data were excluded from the multivariate analysis.

## 3. Results

Notably, 53 out of 65 patients were included in the analysis. A total of 12 were excluded due to the following: 5 were never treated, 4 did not have their blood collected before commencing treatment, and 3 were treated with pembrolizumab alone or chemotherapy alone (Appendix A). Six patients were excluded from the multivariate analysis due to the missing following data (two unknown smoking, three unknown PD-L1 and one unknown NLR). All included patients received pembrolizumab with chemotherapy. The chemotherapy regime included carboplatin in combination with one of the following agents: pemetrexed (34 patients), paclitaxel (18 patients) or vinorelbine (1 patient). Patient demographics included in the analysis are summarised in Table 1. Of note, the majority of patients (76%) either do not express PD-L1 (22/53) or had a TPS of 1–49% (18/53). The frequency of HLA-I supertypes in our cohort is comparable to those presented in other studies [10,11] (Appendix A).

The CBR of the cohort was comparable to the documented rate among NSCLC patients treated with pembrolizumab in combination with chemotherapy [5], with 38% (20/53) of patients experiencing CR or PR. CBR was achieved in 42/53 (79%) patients, which includes patients with SD for 6 months or more. The median follow up was 335 days (31–845 days).

### 3.1. Correlation between HLA-I/II Homozygosity and Clinical Outcome

Our analysis did not find a statistically significant association between homozygosity at one or more HLA-I loci and improved CBR (OR = 0.94, 95%CI 0.24–4.30, *p* = 0.625) or HLA-II loci (OR = 1.14, 95%CI 0.33–4.88, *p* = 0.558). Additionally, we did not find any statistically significant association between zygosity at HLA-I or -II and PFS or OS (Figure 1). No association was found after controlling for potential confounders in the multivariate analysis (Table 2). We observed irAEs in 16/53 (30%) patients, with eight of them developing G3 or more toxicity. No statistically significant association was found between HLA-I (RR = 0.66, 95%CI 0.31–1.54, *p* = 0.256) or HLA-II (RR = 2.23, 95%CI 0.83–6.80, *p* = 0.109) zygosity and the development of irAEs.

### 3.2. Correlation between HLA-I Supertypes and Clinical Outcome

When stratified by HLA supertypes, carrying the HLA-A01 supertype was associated with unfavourable PFS (HR = 2.32, 95%CI 1.13–4.77, *p* = 0.022) and OS (HR = 2.86, 95%CI 1.06–7.70, *p* = 0.038) among patients with locally advanced or metastatic NSCLC treated with pembrolizumab in combination with chemotherapy (Figure 2). The presence of HLA-B27, on the other hand, was associated with improved PFS (HR = 0.35, 95%CI 0.18–0.71, *p* = 0.004) and a trend towards improved OS (HR = 0.53, 95%CI 0.19–1.46, *p* = 0.219). None of the other supertypes showed an association with PFS or OS.

None of the HLA-I supertypes were found to be correlated with the development of irAEs (Figure 3). The presence of the HLA-A03 supertype showed a trend in association with protection against developing irAEs that did not reach statistical significance (RR = 0.35, 95%CI 0.12–0.96, *p* = 0.066). HLA-B58 also showed a trend towards increased risk of development of irAEs that did not reach statistical significance (RR = 3.64, 95%CI 1.17–11.38, *p* = 0.087) as well. None of the other supertypes were found to have a protective or unprotective role in the development of irAEs among NSCLC patients treated with pembrolizumab in combination with chemotherapy.

## 4. Discussion

The herein results did not show a correlation between genomic HLA-I or -II zygosity and clinical outcomes in patients with advanced or metastatic NSCLC treated with chemoimmunotherapy, namely anti-PD1 (pembrolizumab) in combination with chemotherapy. However, our study showed a correlation between HLA-A01, -A03 and -B27 supertypes and clinical outcome among those patients. This is the first study reporting on prognostic value of HLA zygosity and supertypes in this population. While there is evidence that HLA-I homozygosity is associated with shorter survival among NSCLC patients treated with checkpoint inhibitors in the first- or second-line inhibitors [10,11], this association seems to be more pronounced among patients with a PD-L1 expression of 50% or more of cancer cells [10]. Concordant with the current clinical practise, most of the patients in our cohort express PD-L1 in <50% of their cancer cells. This difference in cohort, in addition to the treatment regimens, might explain the absence of the association. Chibber 2022 [17] and Negrao 2019 [18] noted a lack of association between HLA-I zygosity and response to pembrolizumab. However, their findings cannot be compared to our previous findings [10], as they included all tumour types treated with pembrolizumab in their analysis. They are also not comparable to the findings herein, as their studies focused on pembrolizumab monotherapy. Our results might reflect the diminished role of HLA molecules among patients with low or no PD-L1 expression in their cancer cells.

Moreover, our study suggests that the presence of HLA-A01 among patients with advanced NSCLC treated with pembrolizumab in combination with chemotherapy is associated with shorter PFS and OS comparing to those who do not harbour this supertype. HLA-B27 positivity is associated with improved PFS and a trend towards improved OS. Of note, the wide confidence interval for the PFS and OS HR associated with HLA-A01 supertype might reduce its significance comparing to HLA-B27. Other supertypes have been found to be associated with improving or worsening survival across different tumour groups [10,11,15]. The limited overlap between these findings might be due to different epitopes produced by different cancer types and therapeutic modalities, which may cause preferential presentation by different HLA supertypes [19]. The addition of chemotherapy to immunotherapy has the potential to improve tumour peptide presentation by enhancing antigen cross-presentation, leukocyte trafficking and cytokine signalling, hence improving treatment outcomes [20]. However, it is still to be evaluated whether chemotherapy induces epitopes with small aliphatic and aromatic/large hydrophobic amino acids in the B and F pockets, respectively, which are preferentially presented HLA-A01 molecules. Further expansion of the knowledge of supertypes and peptide-binding groove characteristics can help in advancing peptide-based cancer vaccine and treatment personalisation. Multiple clinical trials are treating patients based on the expression of HLA supertypes across different tumour streams [21,22].

Additionally, we found that there is a trend towards protecting against development of irAE among patients with advanced or metastatic NSCLC treated with chemoimmunotherapy who harboured HLA-A03. Interestingly, the same supertype, HLA-A03, was found to be correlated with increased risk of development of irAEs among patients with advanced or metastatic NSCLC treated with single-agent anti-PD1/-PDL1 [12]. This emphasises the role of chemotherapy in modulating epitopes produced by cancer tissue and immune system response when added to immunotherapy [23]. Our study might support the use of chemoimmunotherapy among NSCLC patients with the HLA-A03 supertype instead of single-agent immunotherapy if the development of irAE is of concern. Although 43% of patients in Abed et al. 2022 express PD-L1 in ≥50% of cancer tissue [12] compared to 19% in the present cohort, this might have no influence on our findings, as tumour PD-L1 is not reported to be associated with the development of irAEs. The association trend between HLA-B58 and irAEs might be of less significance given the wide confidence interval observed.

Notably, we did not assess HLA expression within the tumour, which is known to affect response and survival outcomes [8,9,24]. NK cell-mediated killing is another mechanism of fighting malignant cells, but it requires the loss of expression of somatic HLA expression on tumour cells [25]. Immunochemotherapy may induce changes in the tumour microenvironment, such as depletion of myeloid-derived suppressor cells [26] affecting NK cell populations and functions, indirectly reflecting changes in HLA usage.

We must notice some limitations that may affect our conclusions, such as the small number of patients in the cohort that might affect the quality of the multivariable analysis. However, we were able to distinguish differences in OS between HLA-I homozygous and those with maximal heterozygosity in a cohort of 61 patients with advanced NSCLC with PD-L1 ≥ 50% treated with pembrolizumab [10]. Based on those results, our power calculation estimated that 44 subjects would have provided sufficient power (0.8) to distinguish a HR ≥ 2. Another limitation is the heterogeneity of our cohort, as 25% of patients either expressed PD-L1 ≥ 50% or their PD-L1 status was unknown.

## 5. Conclusions

The findings of our study reflect the reduced role of genomic HLA zygosity among NSCLC patients with low or no PD-L1 expression on their cancer cells. This reinforces the need to explore biomarkers to guide clinician decision when treating those patients to identify those who will benefit the most from the combination of chemotherapy with immunotherapy in order to reduce the burden of chemotherapy-related toxicity and use chemotherapy when there is strong evidence to do so. Additionally, certain HLA-I supertypes were found to be associated with variable clinical outcome, including the development of irAEs. Nevertheless, those findings need to be investigated further, and the results need to be reproducible before becoming part of clinical practise.

## Figures and Tables

**Figure 1 cancers-16-03102-f001:**
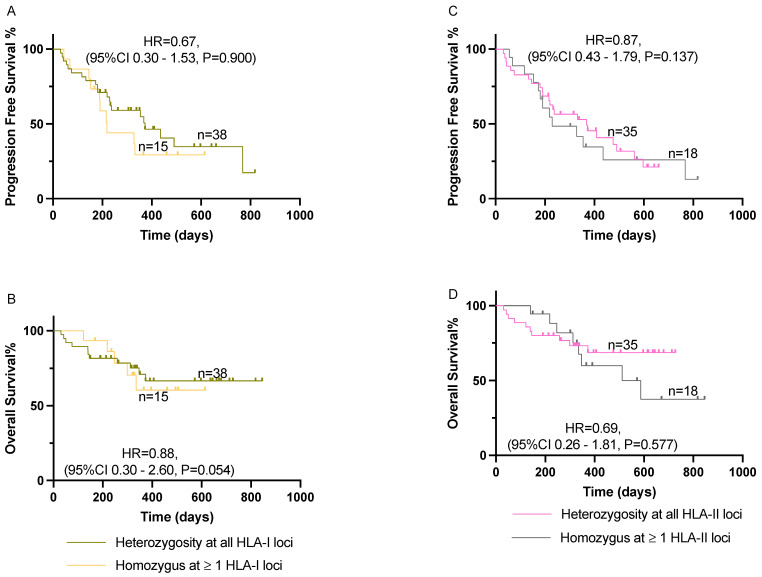
Effect of HLA-I/II homozygosity on PFS and OS among patients with advanced NSCLC treated with pembrolizumab in combination with chemotherapy. (**A**,**B**) Comparison of patients with homozygosity at one or more HLA-I loci on PFS and OS, respectively. (**C**,**D**) Comparison of patients with homozygosity at one or more HLA-II loci on PFS and OS, respectively. Kaplan–Meier curves were compared using log-rank analysis. HLA, human leukocyte antigen; NSCLC, non-small cell lung cancer; OS, overall survival; PFS, progression-free survival.

**Figure 2 cancers-16-03102-f002:**
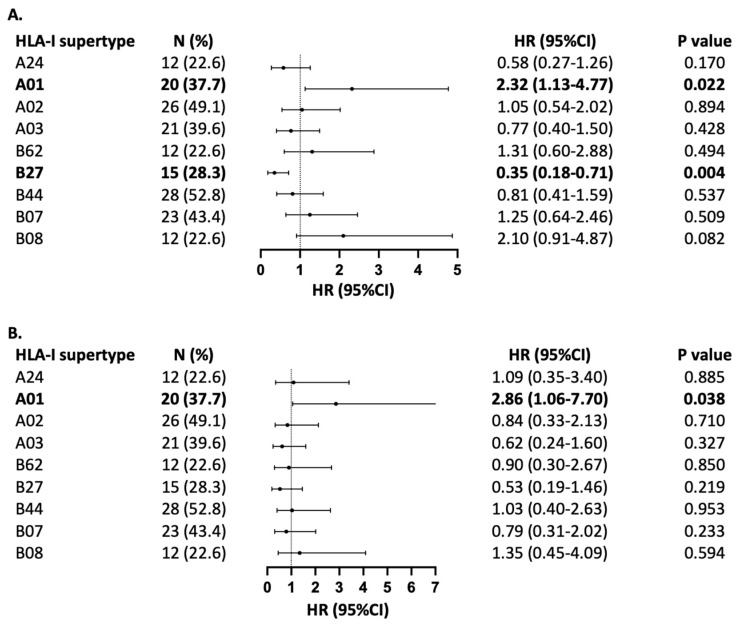
Association of HLA-I supertypes with (**A**) PFS and (**B**) OS. Forest plot indicates hazard ratio (HR) with 95% confidence interval (CI). The number of patients with each supertype is indicated with (N), with % representing frequency in relation to the full cohort of 53 patients with NSCLC treated with pembrochemotherapy. HLA-B58 is not represented in the forest plot, as its frequency was only in two patients.

**Figure 3 cancers-16-03102-f003:**
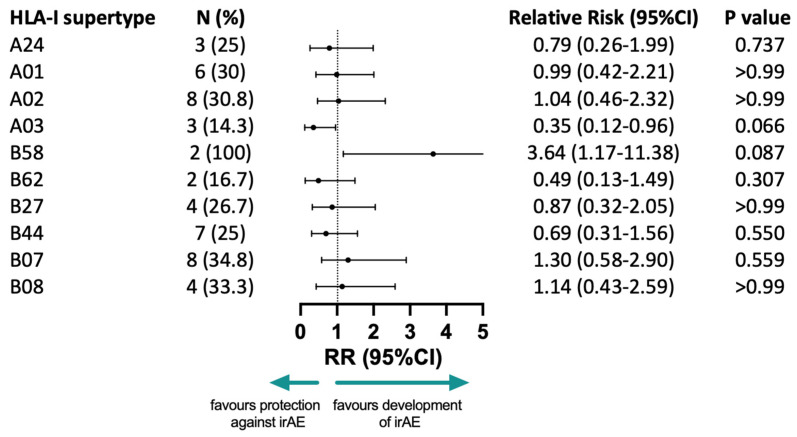
The correlation between HLA-I supertypes and the development of any irAE. N: number of patients who developed irAEs with a specific supertype, with % representing the frequency in relation to the full number of patients that developed irAEs (16 patients); the RR: relative risks, associated CI: confidence intervals and *p*-values of a two-sided Fisher’s exact test are represented; irAE: immune-related adverse event.

**Table 1 cancers-16-03102-t001:** Demographics and Disease Characteristics of Patients at Baseline.

Characteristic	N (%)
**Age**	
≥65	39 (70)
<65	14 (30)
**Sex**	
M	34 (64)
F	18 (34)
**ECOG**	
≤1	47 (89)
>1	6 (11)
**Smoking**	
Yes	47 (89)
No	4 (7)
Unknown	2 (4)
**Stage**	
II	0
III	8 (15)
IV	45 (85)
**Histopathology**	
Adenocarcinoma	38 (72)
SCC	13 (24)
Others	2 (4)
**Molecular status #**	
KRAS mutant	14 (35)
KRAS wild type	22 (55)
KRAS unknown	3 (7)
EGFR, ALK or ROS-1 mutant	1 (3)
**PDL1 expression**	
≥50%	10 (19)
1–49%	18 (34)
0%	22 (42)
Unknown	3 (6)
**Genomic HLA-I zygosity**	
Homozygous at ≥1 loci	15 (28)
Heterozygous at all loci	38 (72)
**Genomic HLA-II zygosity**	
Homozygous at ≥1 loci	18 (34)
Heterozygous at all loci	35 (66)
**Total**	53

*p* value represents the statistical difference between the two cohorts. # Molecular status was only examined in NSCLC with non-squamous cell carcinoma histology (40 patients), ALK: echinoderm microtubule-associated protein like-4-anaplastic lymphoma kinase (EML4/ALK) fusion, ECOG: Eastern Cooperative Oncology Group, EGFR: epidermal growth factor receptor, HLA-I/-II: human leukocyte antigene-I/-II, KRAS: Kirsten Rat Sarcoma GTPase, NSCLC: non-small cell lung cancer, PDL1: programmed death ligand, ROS-1: echinoderm c-ros oncogene 1 fusion, SCC: squamous cell carcinoma.

**Table 2 cancers-16-03102-t002:** Multivariate analysis for the association between HLA-I zygosity and OS among patients with NSCLC treated with pembrolizumab in combination with chemotherapy (N = 47).

	Univariate Analysis				Multivariate Analysis			
Variable	*p*-Value	HR	95.0% CI		*p*-Value	HR	95.0% CI	
			Lower	Upper			Lower	Upper
HLA-I (Hetero vs. Homo)	0.817	1.125	0.415	3.054	0.486	0.679	0.228	2.018
PD-L1 (≥50 vs. <50)	0.364	1.275	0.754	2.154	0.487	1.261	0.656	2.423
Sex (F vs. M)	0.359	0.766	0.434	1.353	0.202	0.668	0.359	1.242
**Age** (≥65 vs. <65)	**0.004**	**2.023**	**1.245**	**3.287**	**0.008**	**2.045**	**1.208**	**3.462**
Smoking (N vs. Y)	0.417	4.842	0.107	218.575	0.986	500.561	0.000	INF
ECOG (≤1 vs ≥2)	0.283	0.706	0.374	1.333	0.357	0.712	0.345	1.468
NLR (<5 vs. ≥5)	0.189	0.722	0.444	1.174	0.118	0.628	0.350	1.125

CI: confidence interval, ECOG: Eastern Cooperative Oncology Group, F: female, HLA: human leukocyte antigen, HR: hazard ratio, M: male, N: no, NLR: neutrophil/lymphocyte ratio, NSCLC: non-small cell lung cancer, OS: overall survival, PD1/PDL1: programmed cell death protein-1/programmed death ligand 1, Y: yes. Bold text represents statistically significant results.

## Data Availability

The original contributions presented in the study are included in the article/Appendix A; further inquiries can be directed to the corresponding author/s.

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
