# Peer review of "HLA-A01 and HLA-B27 Supertypes, but Not HLA Homozygocity, Correlate with Clinical Outcome among Patients with Non-Small Cell Lung Cancer Treated with Pembrolizumab in Combination with Chemotherapy"

_cancers, 2024, doi:10.3390/cancers16173102_

Round 1

Reviewer 1 Report (Previous Reviewer 3)

Comments and Suggestions for Authors

 have no further comments, the authors have rightly  answered to my questions 

Author Response

have no further comments, the authors have rightly  answered to my questions 

Thanks

Reviewer 2 Report (Previous Reviewer 2)

Comments and Suggestions for Authors

While the revised manuscript has addressed my concerns, there is one statement added to it, "Somatic loss of HLA expression in tumour cells requires escape from NK cell-mediated killing [26]." which is incorrect and ridiculous. It must be corrected.

Author Response

While the revised manuscript has addressed my concerns, there is one statement added to it, "Somatic loss of HLA expression in tumour cells requires escape from NK cell-mediated killing [26]." which is incorrect and ridiculous. It must be corrected.

Response:

We have reword the sentence as (page 8, line 261)

"NK cell-mediated killing is another mechanism of fighting malignant cells but it requires the loss of expression of somatic HLA expression on tumour cells [26]."

This manuscript is a resubmission of an earlier submission. The following is a list of the peer review reports and author responses from that submission.

Round 1

Reviewer 1 Report

Comments and Suggestions for Authors

Authors' clarification of these points as below would be greatly appreciated.  
1. Enlarge sample size is recommended. Small sample size may result in the statistical bias. Previous study indicated that HLA-A02 and HLA-B62 showed significant association with short PFS in lung cancer with larger sample size, but these significance has not been seen in this study. 

2. Could author re-check the result of forest plot in Figure 2 and 3?
It's noted that a 95% confidence interval crossing 1 generally implies an insignificant p-value, while not crossing 1 usually indicates a significant p-value. In Figure 2, the overall survival (OS)analysis shows a significant association with the HLA-A01 supertype (p=0.038). However, in the Figure 3, the HLA-B58 supertype appears to show an insignificant result (p=0.087) concerning irAE. 

Comments on the Quality of English Language

The authors have effectively conveyed their concepts in a manner that is both accessible and engaging to readers. 

Reviewer 2 Report

Comments and Suggestions for Authors

The authors investigated potential associations between HLA and response to immune checkpoint immunotherapy (ICI) in NSCLC patients. The major problem is the sole consideration of HLA genes in these association studies. One of the main aspects of HLA is to act as ligands for functionally important receptors for Natural Killer (NK) cells. The authors do not take into consideration this aspect even in their discussion. Other than CTL, NK cells also respond to ICI immunotherapy. Another issue is that the authors provide no rational biological explanation for their observed associations. 

Reviewer 3 Report

Comments and Suggestions for Authors

The paper investigated the association between genomic human leukocyte antigen (HLA) zygosity, HLA supertypes, and clinical outcomes in patients with advanced NSCLC treated with pembrolizumab in combination with chemotherapy. The study prospectively included 53 NSCLC patients receiving this combination treatment. Overall, this study interesting but need some corrections.

The statistical analysis should be more detailed especially for multivariate cox regression model and for association analysis. Did the authors used the two tailed or one tailed p value for Fisher's Exact test???  Did they apply corrections for the p-value for multiple analysis?

The authors shout take care in interpretation of the association, especially between HLA-A01 and OS. The p-value, may be not significant if the corrections are applied, as the lower value of the 95%CI  is less than 1???

In the abstract, there seems to be a typo in line 31 - it should read " HR=2.09, 95%CI 1.03-4.27, P=0.022 " instead of "P=0.22".

The introduction could provide a bit more context on why HLA genomic variations may impact immunotherapy response and toxicity.

In method the number of patients should be added in the text

Consider including a limitations section to discuss potential study limitations, such as the relatively small sample size.

Comments on the Quality of English Language

Needs polishing